# Blocking HIF to Enhance NK Cells: Hints for New Anti-Tumor Therapeutic Strategies?

**DOI:** 10.3390/vaccines9101144

**Published:** 2021-10-07

**Authors:** Massimo Vitale, Monica Parodi

**Affiliations:** UO Immunologia, IRCCS Ospedale Policlinico San Martino Genova, 16132 Genoa, Italy; monica.parodi@hsanmartino.it

**Keywords:** NK cells, tumors, HIF

## Abstract

Natural Killer (NK) cells are becoming an ever more promising tool to design new anti-tumor strategies. However, two major issues are still a challenge to obtain versatile and effective NK-based therapies: the way to maximize the persistency of powerful NK effectors in the patient, and the way to overcome the multiple escape mechanisms that keep away or suppress NK cells at the tumor site. In this regard, targeting the hypoxia-inducible factors (HIFs), which is important for both tumor progression and immune suppression, may be an opportunity. Especially, in the context of the ongoing studies focused on more effective NK-based therapeutic products.

## 1. NK Cells as Anti-Tumor Effectors

The continuous studies on NK cells have enormously amplified the knowledge on their development, body distribution, and functions, laying the bases for their effective use in immunotherapy against tumors. It is well defined that human NK cells can kill tumor targets by the use of major activating surface receptors (i.e., NKp46, NKp30, NKp44, NKG2D, DNAM1), which recognize antigens overexpressed by tumor cells, and avoid attacking normal cells by means of inhibitory receptors (KIRs, NKG2A, and LILRB1) recognizing HLA class I molecules (whose expression is indeed often downregulated in tumor cells) [1,2]. It is also quite well known that, during cell-to-cell contact, the different receptor types could be engaged, coordinatively with the LFA1 adhesion molecules, to form inhibitory or cytotoxic immunological synapses, the latter resulting in the delivery of perforins and pro-apoptotic granzymes into the tumor target cell [3]. The need for a tight control of the NK cell cytotoxicity has led, in humans, to the generation of a quite complex pattern of inhibitory receptors each endowed with peculiar specificity: NKG2A and LILRB1 (also known as ILT2 or LIR1) recognizing respectively HLA-E, and (predominantly) HLA-G molecules, and each member of the quite large family of KIRs recognizing a different group of HLA class I alleles [2,4,5]. To further complicate matters, NK cells also express activating homologues of HLA-specific receptors (i.e., NKG2C and activating KIRs), which are involved in the response against virally infected cells. The HLA-specific receptors (both inhibitory and activating) are not homogeneously expressed on the whole NK cell population of each individual, rather they are dispersed, giving rise to a complex repertoire. Perturbations of such a repertoire could be associated with certain viral infections (in particular CMV) indicating possible expansion and persistence of virus-responsive “memory-like” (also defined as “adaptive”) NK cells in seropositive individuals [6,7,8]. Strikingly, in adaptive NK cells predominates the CD16-mediated activating pathway leading to effective target cell killing via ADCC [7,8]. Therefore, adaptive NK cells may be particularly suitable to complement the curative effects of therapeutic abs to tumor Ags. Nevertheless, these NK cells may also have a direct anti-tumor effect, independent of the targeting abs. Indeed, adaptive NKG2C^+^ NK cells have been shown to expand specifically in response to CMV reactivation in recipients of hematopoietic cell transplants (HCT) and to play a role in the control of leukemia relapse [9].

By virtue of the partial HLA class I mismatch between donor and recipient in haploidentical HCT or NK cell transfer, it is possible that a fraction of donor-derived NK cells could express inhibitory KIRs not recognizing any recipient’s HLA class I molecule [10]. These cells do not appear to cause serious negative effects in the recipient, probably because most normal cells poorly express the ligands for activating NK receptors and therefore are hardly attacked by NK cells even in the absence of inhibitory signals. Instead, a major problem for an effective use of donor-derived NK cells in immunotherapy may be to maintain the immunological competence of allogeneic NK cells in the recipient. NK cells generally acquire their cytotoxic capabilities through a process of “education” [11], which proceeds alongside the latter phases of maturation. These phases are marked by the acquisition of NKG2A expression first, and then, by the replacement of NKG2A with the more potent inhibitors, KIRs. The education process enables NK cells to increase their cytotoxic potential proportionally to the strength of the inhibitory signals received by recognition of “self HLA class I”. Therefore, NK cells expressing KIRs not recognizing self HLA molecules will not acquire cytotoxicity, while those expressing NKG2A will show intermediate cytotoxic potential, and those expressing KIR specific for “self” will show maximal cytotoxic potential. NK cells are thought to be educated in secondary lymphoid tissues, where NK cell maturation occurs [12], but the process may also take place in the periphery. Intriguingly, further studies in mice have suggested that, actually, mature NK cells can continuously adapt their function to the environmental changes of “self” [13,14,15]. Based on this observation, we can infer that in humans, allogeneic NK cells, once transferred into the patients, may reduce their cytotoxicity in response to the diminished signaling of KIRs (which fail to recognize recipient’s HLA class I alleles); even though, a recent study has proposed that the recognition of self HLA molecules “in cis” (i.e., receptor/ligand interaction on the same cell) can contribute to maintain education [16]. A different situation may occur in NK cells developing after haploidentical hematopoietic stem cell transplantation (HSCT) in patients with high-risk leukemias. In this case, allogeneic NK cells may be fully competent, as educated by the HLA class I alleles expressed by the donor’s transplanted hematopoietic cells that populate the recipient [10]. Another problem to maintain immunological competence is related to the tendency of NK cells (and also T cells, actually) to respond to prolonged activation due to chronic stimuli (such as the exposure to tumor cells and cytokines), by expressing inhibitory checkpoint receptors: TIGIT, TIM-3, and PD-1 [17,18,19]. The ligands for these receptors can also be expressed by tumor cells; therefore, targeting the checkpoint receptor/ligand pairs with specific abs represents a current strategy to unleash the patient’s anti-tumor immune responses.

It is noteworthy that most studies on NK cell-based immunotherapy have so far been carried on hematologic malignancies, while fewer attempts have been made with patients with solid tumors. This depends in part on the different therapeutic strategies currently in use for hematologic and solid malignancies, the latter often involving surgery, but it is also related to additional issues regarding the interactions of NK cells with the peculiar microenvironment distinctive of solid tumors. Indeed, at the tumor site several mechanisms are induced to limit the recruitment of NK cells and also to locally suppress their functions [20]. Many of these mechanisms are related to the abnormal recruitment or induction of suppressive immune cell populations (Tregs, Myeloid Derived suppressor Cells (MDSC), and Tumor Associated Macrophages (TAM)) and alteration of stromal cells (tumor-associated fibroblasts) and of the extracellular matrix (ECM), and appear to be orchestrated by tumor cells. However, it should be considered that the physical changes due to the uncontrolled growth of the tumor tissue, leading to the generation of a hypoxic status, play a pivotal role.

## 2. Role of HIF in the Orchestration of the Host/Tumor Interface

Reduction of O_2_ tension frequently characterizes expanding tumor tissues and strongly influences the local microenvironment, playing an important role in the orchestration of the host/tumor interface and, ultimately, in the tumor progression. HIF-1α protein (and its isoforms, HIF-2α and HIF-3α) represents the master regulator of the response to hypoxia in both normal and tumor tissues [21]. In O_2_ rich compartments, this protein is in large part degraded and eliminated from the cytoplasm of the cells, whereas, under low O_2_ tension it accumulates and is then translocated to the nucleus, where it complexes with HIF-1β to govern the transcription of many hypoxia-related genes bearing “hypoxia-response regulatory elements”.

Several processes, involved in the tumor development, growth, invasion of surrounding tissues, and metastatic spread, can be influenced or supported by hypoxia via HIF proteins [21,22]. Thus, for example, hypoxia stimulates angiogenesis via VEGF induction, modifies cellular metabolism, and may also improve survival of tumor cells by increasing their tendency to skew from apoptosis to autophagy. Moreover, hypoxia favors the epithelial-to-mesenchymal transition (EMT), an epigenetic process of cellular modification that, when applied to tumor cells, it can promote their switch to less differentiated forms, with reduced anchorage capabilities, increased stemness, and increased resistance to radiotherapy and drugs.

Hypoxia also influences the function of immune cells therefore affecting the host immune response to the tumor. For example, under hypoxic conditions, dendritic cells (DCs) upregulate pro-inflammatory cytokines and chemokines, but also undergo defective maturation with reduced ability to induce proper Th1 activation [23,24]. Acidification of the tumor microenvironment (TME), due to hypoxia-related metabolic changes, suppresses T and NK cells [25,26]. Moreover, hypoxia induces various suppressive factors, including TGF-β, IDO, and PGE2 [27]. Finally, HIF1α can favor Tregs [28] and, indirectly, MDSC and TAM accumulation [29,30].

Different studies have been done also on NK cells, showing how hypoxia could significantly dampen the efficacy of this important arm of the anti-tumor immunity. Hypoxia-induced HIF-1α in osteosarcoma cells has been shown to downregulate the expression of MICA [31], a ligand of the activating receptor NKG2D. By converse, hypoxia can upregulate the expression of PD-L1 and HLA-G [32,33], which are part of different checkpoint receptor-ligand pairs (PD-L1/PD-1, HLA-G/LILRB1, HLA-G/LILRB2, HLA-G/KIR2DL4) inhibiting both NK and T cell activity. Another mechanism to escape NK cell attack is represented by the hypoxia-induced autophagy. By this process, tumor cells can inactivate granzyme B proapoptotic molecules that are delivered into their cytoplasm by NK cells during cell-to-cell contact [34]. In general, the hypoxic microenvironment suppresses NK cells in different ways. For example, hypoxic TME induces mitochondrial fragmentation of tumor infiltrating NK cells, limiting their cytotoxicity and anti-tumor activity [35]. Moreover, the accumulation of Tregs, MDSC, immature DC, and suppressive factors affect NK cells [20]. In this regard, microvescicles released by hypoxic tumor cells have been shown to deliver to NK cells TGF-β and miRNA-23a, which can reduce the expression of NKG2D and of the lytic granule-associated CD107a molecule, respectively [36]. miRNA23a has also been shown to inhibit cathepsin C expression, resulting in granzyme B activity reduction [37]. In addition, HIF typically up-regulate expression of ecto-5′-nucleotidase (CD73) broadly in the cells of the TME, including stromal cells, tumor cells, endothelial cells, Tregs, and DC. CD73 hydrolyzes AMP to adenosine, which acts on NK cells by inhibiting cytotoxicity and cytokine expression [38]. Strikingly, also the recruitment of NK cells into the tumor nests can be inhibited by hypoxia via HIF-1α. Indeed, targeting HIF-1α in a melanoma mouse model resulted in increased cytotoxic NK and T cell infiltration in the tumor, which showed increments in CCL2 and CCL5 chemokines [39]. Nevertheless, even more importantly, hypoxia also has a direct effect on NK cells, possibly affecting their metabolism and function [26]. Indeed, HIF-1 can orchestrate the expression of genes involved in the energy production, favoring the process of glycolysis and inhibiting oxidative phosphorylation (OXPHOS) [40,41]. At steady state NK cells show relatively low glycolysis and OXPHOS rates, which, however, are significantly increased upon cytokine exposure, to support enhanced cytotoxicity and IFN-γ production [42]. Similarly, adaptive NK cells appear to maintain higher rates of glycolysis and OXPHOS [43]. Therefore, in both cytokine-induced and adaptive NK cells, hypoxia may impact metabolism reprogramming, resulting in reduced OXPHOS and increased glycolysis (poorly efficient when uncoupled to OXPHOS). By contrast, educated NK cells may be only marginally affected. Indeed, their metabolic advantage on uneducated cells would depend on higher GLUT1 glucose receptor expression and higher glycolysis, rather than on OXPHOS [44]. Metabolic alterations can contribute in part to the hypoxia related modulation of different NK cell functions. It has been demonstrated that, under low O_2_ tension, NK cells change their mode to respond to cytokines, such as IL-2 or IL-15 [45,46]. For example, they lose their ability to upregulate the expression of key activating receptors involved in target cell recognition, failing to improve their tumor cell killing capabilities [45]. They also reduce the release of cytokines and chemokines (IFN-γ, GM-CSF, CCL3, and CCL5) in response to the combined stimulus of IL15+IL18, or improve the ability to migrate in response to specific chemokines, such as CCL19, CCL21, and CXCL12 [41]. Remarkably, this latter effect appears to be restricted to the CD56^bright^CD16^dim^ NK cell subset, suggesting that this NK cell population could more easily infiltrate hypoxic tissues expressing such chemokines. The transcriptomic analysis of NK cells exposed to low O_2_ tension reveals a profound modification of many biological processes, spanning from those related to metabolism to those involved in more specific functions, including cytokine release and regulation of cytolytic response [41,46]. The large number of hypoxia-modulated genes (HMGs) and the Gene Set Enrichment Analysis indicates an actual adaptation of NK cells to hypoxia, with the involvement of HIF-1α and HIF-2α [41]. This latter point is further confirmed by an important study showing how the specific inhibition of HIF-1α expression on NK cells can deeply change their behavior in the tumor hypoxic environment. Ni et al., using a mouse model with the conditional deletion of HIF-1α in NK cells, elegantly demonstrated that lack of HIF-1α renders certain NK cells capable of producing IFN-γ in response to IL-18 in the tumor hypoxic environment and that these HIF-1α^−/−^ NK cells can infiltrate and control the tumor in vivo [47].

## 3. HIF Inhibitors and NK Cells

On the whole, the studies on hypoxia in the TME indicate HIF molecules, in particular HIF-1α, as key players for both the tumor progression and the escape from the NK cell attack (and more widely, from the immune mediated control of the tumor). Therefore, several therapeutic agents targeting HIF molecules have been considered, and are currently analyzed in phase II/III clinical trials [21]. Such agents include molecules that can inhibit HIF protein synthesis or mRNA expression, induce non-functional HIF dimers, or induce HIF degradation. Clinical studies suggest some benefit for some of these molecules, encouraging their further evaluation as single agents or in combined therapies. In this regard, it is desirable that effective new strategies, combining HIF-inhibitors to NK-based immunotherapy, could be conceived soon. In this case, however, several questions should be examined in the attempt to maximize the therapeutic success, especially considering the growing complexity of the NK cell biology revealed by more recent studies (Figure 1). One point may be which NK cell subset could benefit more from the HIF inhibition. In this regard, the study by Ni et al. suggests a certain variability within mice NK cells in the response to the lack of HIF, indeed, only a fraction of tumor infiltrating HIF^−/−^ NK cells were activated in the studied model. In humans, it could be interesting to assess whether the HIF response could vary in NK cells depending on their stage of terminal differentiation (i.e., characterized by different expression levels of CD56, CD16, NKG2A, KIR, and CD57), on their educational status, or even on their residency in different body compartments. Regarding this latter point, it should be considered that different organs also include tissue-resident NK cells with distinctive functions, such as uterine NK cells, which are mostly involved in supporting embryo development, or liver NK cells, facing viral infections via TRAIL [48]. The O_2_ tension significantly varies in different organs and tissues [49]; therefore, HIF response may variably affect specialized functions of resident NK cells. Another intriguing question may regard the so-called adaptive NK cells, which are characterized by peculiar phenotype and transcriptional profile and may differently respond to HIF-inducible pathways [8]. It is also important to consider how NK cells are stimulated or prepared for NK-based therapies [10], as it is still poorly known whether preliminary or prolonged activation may change their response to hypoxia. Thus, stimulation of NK cells with IL-2 or IL-15 before infusion, or the subsequent administration of IL-2, IL-15, or the IL-15 superagonist ALT-803 [50] to sustain NK cell activation in the patients, may differently influence the possible effects of HIF inhibitors. It is also unknown the effect of hypoxia on the Cytokine Induced Memory-Like (CIML) NK cells, which are prepared from peripheral blood NK cells after brief stimulation in vitro with a cytokine cocktail of IL-15, IL-12, and IL-18. By studies in vitro and in animal models, CIML-NK cells have been demonstrated to be long-lived and to promptly respond to stimuli, and are now being evaluated in clinics [51,52]. Therefore, their potential use in combination with HIF inhibitors may be contemplated in future studies. Other NK cell preparations that should be evaluated for the possible effects of HIF inhibitors are represented by those derived from umbilical cord blood (UCB) primary NK cells or their precursors (which are already studied in clinics), and from the “induced Pluripotent Stem Cells” (iPSC), both representing promising “off-the-shelf” products for immunotherapy [53,54]. iPSCs are induced from somatic cells that are transfected with reprogramming factors and then cloned, expanded, and stored in biobanks to be used for the generation of the desired differentiated somatic cells, including NK cells. Compared to traditional NK cell products, the iPS-NK may present the advantage to provide higher numbers of cells (which are required for therapeutic protocols) and to be more homogeneous and genetically editable [55]. Of note, both UCB and iPSC represent reliable platforms for the engineering of NK cells to generate Chimeric Ag Receptor (CAR)-NK cells. Intriguingly, the manipulation of CAR-NK cells may consider targeting HIF-1a in order to get empowered effector cells. Finally, a further question regards the possible combination of HIF inhibitors with additional boosts that can redirect NK cells to target tumor cells. This is the case of therapeutic abs to tumor Ag, which can induce ADCC, or bi- and tri-specific Killer Engagers (BiKE and TRiKE), engineered molecules simultaneously targeting tumor Ag, engaging CD16 on NK cells, and (in the case of TRiKE) presenting IL-15 to support NK cell persistence and expansion [10,56]. Both ADCC and BiKE/TRiKE stimulation are mediated by CD16, which has been shown to be poorly affected by hypoxia [45]. In this case, HIF inhibitors may have only marginal effects; however, new promising NK engagers have been designed to trigger different activating receptors—such as NKp46, NKp30, and NKG2D [57]—which, on the contrary, have been shown to be significantly suppressed in hypoxic conditions.

In conclusion, NK-based immunotherapy is a rapidly evolving area of research, which is prevalently focused on the way to obtain NK cell products that are ever more powerful and precise in targeting tumors. These cells should also respond to requirements of durability in the patient, have easy availability, and be capable of facing the suppressive mechanisms mounted by tumor tissues. In this latter aspect, the study of how to circumvent the effects of hypoxia, and specifically of its master regulator HIF complex, may offer future therapeutic synergies.

## Figures and Tables

**Figure 1 vaccines-09-01144-f001:**
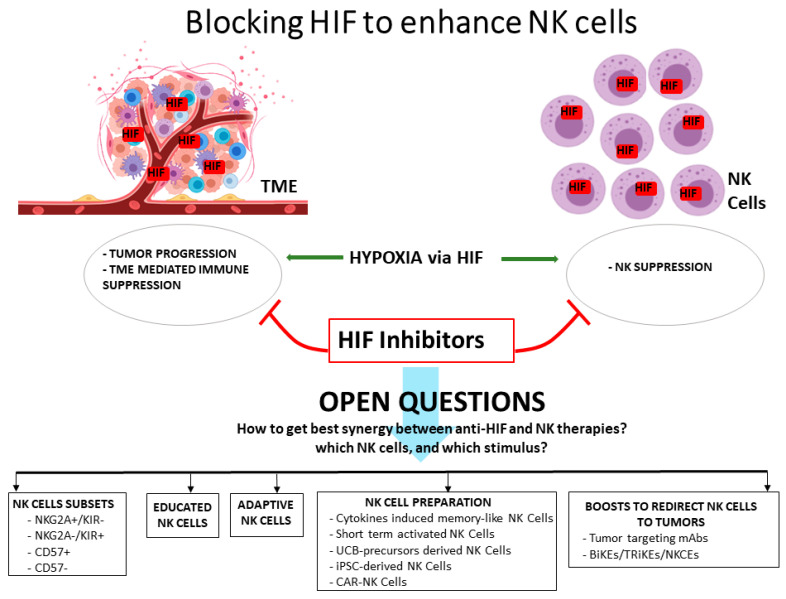
Blocking HIF to enhance NK cells.

## Data Availability

No raw data have been produced for the preparation of this article.

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
