# Peer review of "Blocking HIF to Enhance NK Cells: Hints for New Anti-Tumor Therapeutic Strategies?"

_vaccines, 2021, doi:10.3390/vaccines9101144_

Round 1

Reviewer 1 Report

Overall an interesting overview of the topic. I suggest a few improvements.

I am not sure about the format requirement; I think a very simple figure could help present the main messages. Perhaps they could use the title of their subtitles 1, 2, 3, suggest some directions to take, state clear or unsolved aspects. I think anything would help at this point. And could be used as well graphical abstract

L38: authors should define “adaptive” NK on first appearance

L55 or where most relevant, Authors could briefly mention where NK-cell education occurs in general or reoccurs.

L73: “by expressing inhibitory checkpoint receptors: TIGIT, TIM-3 and PD1 (10)” here authors should try to include addition references (not review) from significant work focusing on NK and these immune checkpoints

L31 HLA class I alleles (4,5,2). To further complicate matters, NK cells also express activating Verify police size

L93 HIF-2a and HIF-3a. I would suggest to use symbols for alpha and beta “throughout” the manuscript

L11: same remark for TGF-beta or with the symbol looks better, but not TGFb is not appropriate

L121: PD-L1 and PD-1 should be preferred. PDL-1/PD1 looks inappropriate. Please verify through

Regarding ref36: I think there is an error on first name/surname of the first author. Please double check. This should be Ni J et al as well mentioned L153

  1. Jing, N.; Xi Wang, A.; Stojanovic, Q.Z.; Wincher, M.; Bühler, L.; Arnold, A.; P. Correira, M.; Winkler, M.; Koch, P.S.; Sexl, V.; 328 Höfer, T.; Cerwenka, A. Single-Cell RNA Sequencing of Tumor-Infiltrating NK Cells Reveals that Inhibition of Transcription 329 Factor HIF-1α Unleashes NK Cell Activity. Immunity. 2020 Jun 16;52, 1075–1087.e8. DOI: 10.1016/j.immuni.2020.05.001.

references part:  15) 15) repetition to be corrected

Author Response

We thank this reviewer for the positive judgement of the manuscript, and, especially, for her/his ameliorative suggestions and notes. We have replied to all the questions as listed below.

Point-by-point reply.

1) “… a very simple figure could help present the main message.”

We actually also included a Figure file in the first submission, which, evidently, didn’t reach the Referees. We guess that some technical problems prevented its correct saving within the submission platform.

The figure, indeed, covered the issue proposed by the Referee. We have now slightly modified it to further adapt to the Referee’s indications.

2) “L38”.

The “adaptive” NK cells have been defined.

3) “L55” “…Authors could briefly mention where NK-cell education occurs in general or reoccurs.

We thank the reviewer for having stressed this point. To our knowledge, the sites where NK cells are educated, or modulate their functions, have not been completely defined yet. Bone marrow may be a place, or more likely, the secondary lymphoid tissues (i.e. where NK cells should undergo latter maturation steps). However, this process may also occur in the periphery, since also mature NK cells can modify their activity in response to changes of MHC-class I expression. Finally, the education “in cis” (i.e. through the interaction of KIR and HLA molecules expressed on the surface of the same NK cell) has also been proposed. Two sentences on these points have been added in the revised version of the manuscript (lines 66-67 and 72-74) (65-66 and 71-73 in the version without track changes).

4) “L73”.

Refs on TIGIT, TIM3 and PD-1 have been added.

5) “L31”.

The character size has been adjusted.

6) “L93”.

“HIF-2a” and “HIF-3a" have been inserted properly.

7) “L11”.

“TGF-b” has been inserted properly.

8) “L121”.

“PD-L1” and “PD-1” have been inserted properly.

9) “mistakes in Refs 36 and 5”

Thanks for having pointed them out. We have amended the errors.

During the revision we have also inserted additional minimal changes to correct little errors and, in general, to improve the text.

Reviewer 2 Report

In this view point, the authors discuss the current knowledge and future perspective in use NK cells in cancer therapy with a particular focus on hypoxia and solid tumors. Despite some of the topic discussed in this manuscript have been recently discussed in different reviews, this manuscript discusses many aspects from a novel point of view representing an interesting reading.

I encourage to improve same points, including improving references. The suggested references represent only examples; the authors are free to include different reference to discuss the topics:

-Line 55: what the authors means by "parallels"? Please rephrase this sentence.

-Line 73: I suggest to mention also CTLA4. Please support this aspect with references including for example (not necessarily):
1. Lanuza PM, Pesini C, Arias MA, Calvo C, Ramirez-Labrada A, Pardo J. Recalling the Biological Significance of Immune Checkpoints on NK Cells: A Chance to Overcome LAG3, PD1, and CTLA4 Inhibitory Pathways by Adoptive NK Cell Transfer? Front Immunol. 2020 Jan 9;10:3010. doi: 10.3389/fimmu.2019.03010. PMID: 31998304; PMCID: PMC6962251.

2. Passariello M, Vetrei C, Sasso E, Froechlich G, Gentile C, D'Alise AM, Zambrano N, Scarselli E, Nicosia A, De Lorenzo C. Isolation of Two Novel Human Anti-CTLA-4 mAbs with Intriguing Biological Properties on Tumor and NK Cells. Cancers (Basel). 2020 Aug 6;12(8):2204. doi: 10.3390/cancers12082204. PMID: 32781690; PMCID: PMC7464132.

3. Sanseviero E, O'Brien EM, Karras JR, Shabaneh TB, Aksoy BA, Xu W, Zheng C, Yin X, Xu X, Karakousis GC, Amaravadi RK, Nam B, Turk MJ, Hammerbacher J, Rubinstein MP, Schuchter LM, Mitchell TC, Liu Q, Stone EL. Anti-CTLA-4 Activates Intratumoral NK Cells and Combined with IL15/IL15Rα Complexes Enhances Tumor Control. Cancer Immunol Res. 2019 Aug;7(8):1371-1380. doi: 10.1158/2326-6066.CIR-18-0386. Epub 2019 Jun 25. PMID: 31239316; PMCID: PMC6956982.

-Line 98: complete the sentence :"to govern the transcription of many hypoxia-related genes" with "bearing hypoxia responsive regulatory elements".

-Line 148: This is a very interesting point. I encourage the authors to detail the effect of Hypoxia on NK cell immunometabolism (metabolic switch etc..). It would be also important to include the effect of hypoxia-mediated adenosine on NK cells. Supporting literature:

1. Terrén I, Orrantia A, Vitallé J, Zenarruzabeitia O, Borrego F. NK Cell Metabolism and Tumor Microenvironment. Front Immunol. 2019 Sep 24;10:2278. doi: 10.3389/fimmu.2019.02278. PMID: 31616440; PMCID: PMC6769035.

2. Choi, C., Finlay, D.K. Optimising NK cell metabolism to increase the efficacy of cancer immunotherapy. Stem Cell Res Ther 12, 320 (2021). https://doi.org/10.1186/s13287-021-02377-8

3. Chambers AM, Matosevic S. Immunometabolic Dysfunction of Natural Killer Cells Mediated by the Hypoxia-CD73 Axis in Solid Tumors. Front Mol Biosci. 2019 Jul 24;6:60. doi: 10.3389/fmolb.2019.00060. PMID: 31396523; PMCID: PMC6668567.

-Line 169: Figure 1 is not attached. Please include figure in the new version of the manuscript.

Author Response

We thank this reviewer for having considered with interest the manuscript. We also appreciate her/his suggestions to improve the article. We have taken into consideration the indications as described below.

Point-by-point reply.

1) “Line 55” “….Please rephrase this sentence”.

We apologize for the unclear sentence. We have now modified the text and hope to have improved it.

2) “Line 73” “I suggest to mention also CTLA4…..”

We thank the Reviewer for the suggestion, however, we feel there is not enough consensus on how CTLA4 can be effectively expressed and work on NK cells in humans (as also indicated in the first suggested Review – Lanuza et al.). Rather, it appears established the inhibitory effect of anti-CTLA4 abs on tumor cells or Tregs, which can be a direct effect of the abs or, indeed, it can be mediated by NK cells via ADCC. An organic discussion on this point, although interesting, is beyond the purpose of our manuscript, therefore, we preferred to maintain the original sentence.

3) “Line 98” “complete the sentence…. with "bearing hypoxia responsive regulatory elements".

We have completed accordingly.

4) “Line 148” “I encourage the authors to detail the effect of Hypoxia on NK cell immunometabolism (metabolic switch etc..). It would be also important to include the effect of hypoxia-mediated adenosine on NK cells”.

We agree with the Referee, the role of hypoxia (and HIF) on NK cell metabolism is indeed an important point. We have now inserted new sentences describing and discussing this issue (lines 150-161) (148-159 in the manuscript version without track changes). Also the description of the immune-modulatory effects of CD73-induced adenosine has been added (lines 143-146). The suggested references (plus additional 3) have been quoted (we thank the Reviewer for the provided supporting literature).

5) “Line 169” “Figure 1 is not attached. Please include figure in the new version of the manuscript”.

We apologize for the inconvenient, the system probably failed to save the submitted figure file. We have now added the figure.

During the revision we have also inserted additional minimal changes to correct little errors and, in general, to improve the text.